# Impact of a multilevel HIV prevention programme on adolescent girls and young women's empowerment for risk reduction: A mixed methods evaluation in Kenya

Elizabeth Kemigisha[1]*, Jane Osindo[1], Vivienne Kamire[2], Venetia Baker[3], Annabelle Gourlay[3], Stephen Gakuo[1], Sarah Mulwa[3], Faith Magut[3], Thomas Gachie[3], Moses Otieno[2], Sammy Khagayi[2], Daniel Kwaro[2], Sian Floyd[3], Abdhalah Ziraba[1], Isolde Birdthistle[3]

1 African Population and Health Research Center, Nairobi, Kenya, 2 Centre for Global Health Research, Kenya Medical Research Institute, Kisumu, Kenya, 3 Faculty of Epidemiology and Population Health, London School of Hygiene & Tropical Medicine, Keppel Street, London, United Kingdom

* ekemigisha@aphrc.org

## Abstract

Empowerment is key to person-centered HIV prevention. This study aims to evaluate the extent to which the Determined, Resilient, Empowered, AIDS-free, Mentored, Safe (DREAMS) Partnership promotes empowerment of adolescent girls and young women (AGYW) in Kenya. We conducted a mixed-methods study to examine DREAMS influence on empowerment precursors (resources, agency, institutional structures) and outcomes, guided by Kabeer's framework for women's empowerment. With representative cohorts of AGYW aged 13–22, followed since 2017/18, we estimated the impact of DREAMS on social support and self-efficacy by 2022. With a nested sample of cohort participants in 2022, we conducted qualitative inquiry through in-depth interviews and participatory action learning with DREAMS participants, implementers, and community members, for evidence of mechanisms, achievement and contextual influences on AGYW empowerment. We found statistical evidence of a causal impact of DREAMS on social support and self-efficacy among adolescent girls. Qualitatively, DREAMS enhanced AGYW's access to and control of resources, including coveted material assets like educational subsidies, hygiene products, and business support, which reportedly reduced transactional sex through economic autonomy and bodily integrity. DREAMS enhanced intangible resources like HIV knowledge and financial skills. Greater confidence, courage, and critical consciousness enabled some AGYW to act on health choices, assume leadership roles, and continue education. In mentor-led activities, AGYW demonstrated collective action and solidarity and group saving skills for financial autonomy. DREAMS aimed to engage the broader community including leaders, parents, and young men, but social structures, dominant gender norms, gender-based violence, and widespread

**Data availability statement:** All relevant data are included in the paper and its supporting information files. To protect the confidentiality of qualitative study participants, the authors will consider requests made to them directly for access to anonymised extracts from interview transcripts. Quantitative datasets will be available at APHRC's Microdata Portal https://microdataportal.aphrc.org/index.php/catalog.

**Funding:** This work was supported by the Bill and Melinda Gates Foundation INV-027727 to AZ. The funders had no role in study design, data collection, and analysis, the decision to publish, or the preparation of the manuscript.

**Competing interests:** The authors have declared that no competing interests exist.

poverty mitigated its empowering impact for AGYW. We found compelling evidence that DREAMS strengthened many young women's access to and control of resources and their intrinsic and collective agency. These are important precursors to empowerment but insufficient for the transformation of power relations without supportive institutional structures, including gender-equitable norms and viable livelihood strategies.

## Introduction

In settings most affected by HIV and AIDS, gender gaps in HIV risk emerge from early adolescence [1,2]. Despite considerable declines in HIV incidence since anti-retroviral treatment availability, adolescent girls and young women (AGYW) experience much higher HIV risk than their male peers. In Kenya, for example, new HIV infections have more than halved since 2013, yet incidence among 15–24-year-old females is more than three times higher than male counterparts, and 10 times higher for 15–19-year-olds [3].

The gendered dimension of HIV risk reflects wider inequalities that impede young women's health, development, and self-determination. From the onset of sexual development, adolescent girls' vulnerability to HIV is shaped by norms wherein women are deferential to men, masculinity is partly defined by intergenerational sex and multiple sexual partners, and violence against women and girls is pervasive [4,5]. The vulnerability of young adolescent girls reflects age and power differentials with male partners, typically older, and the limited support from social structures and institutions for healthy adolescent sexual development [6]. Adolescent girls in Kenya describe shame, fear of judgement and unsupportive parents and healthcare workers as barriers to seeking information and services to protect their sexual and reproductive health (SRH) [7]. Thus, despite the epidemiological progress enabled by biomedical HIV breakthroughs, inequalities in HIV risk persist because they are rooted in entrenched patriarchal systems.

For many years, HIV prevention efforts focused on individual risk determinants, including a focus on adolescents' knowledge and behaviour change. Recognising the limitations of this approach, interventions to influence broader structural determinants of HIV risk have gained importance over the last decade [8]. The World Health Organization has advocated for consideration of social determinants of health, defined as "conditions in which people are born, grow, work, live and age and these may be influenced by a wide set of systems and forces" [9]. Examples include social, cultural, economic and political policies and systems that shape the distribution of material and symbolic power, access and control of resources [10].

Growing appreciation of structural determinants of HIV risk has translated into broader, more comprehensive interventions addressing multiple levels of influence on HIV transmission and acquisition. A recent review of adolescent HIV and SRH interventions shows that programmes have become more explicit in addressing gender inequalities, for example, by addressing contextual determinants of vulnerability, like systemic economic, educational, and social inequities for meaningful impact

[11]. There is no consistent approach to programmes that aim to be 'gender-transformative' by examining and changing harmful gender and power imbalances [12]. So far, examples have adopted diverse strategies to offer interventions across levels of the socio-ecological model, guided by context-specific drivers and a range of theoretical frameworks, e.g., person-centered or rights-based or women's empowerment theories [13,14]. A prominent example of multi-level programming to prevent HIV risk among adolescent girls and young women is the DREAMS Partnership (Determined, Resilient, Empowered, AIDS-free, Mentored, and Safe). Guided by pathways to women's empowerment articulated by Kabeer [15] and extended by the Bill & Melinda Gates Foundation [16], we sought evidence of DREAMS influence on pre-conditions of empowerment. We also sought evidence of young women's achievement (outcomes) of empowerment in areas related to sexual and reproductive health, particularly the expansion of choice, voice, and opportunity.

## Methods

### Evaluation research contexts: DREAMS in Kenya

The DREAMS core package was rolled out in urban informal settlements in Nairobi (Korogocho and Viwandani) and rural Gem in Siaya County, western Kenya from 2016 with PrEP interventions incorporated in 2019. Before onset of DREAMS programs in 2015, over half (51%) of all new HIV infections occurred among adolescents and young people (aged 15–24 years), and 33% among young women [17]. According to the HIV estimates report in 2018, at least five counties (including Siaya and Nairobi) contributed 43% of the estimated total new HIV infections in Kenya, with more than 1000 new infections among young people aged 15–24 years in each of these counties [18]. A review of the policy framework for enabling access to HIV prevention for AGYW including PrEP in Kenya, Uganda and South Africa identified key barriers in existing policies that may hinder access to these services including need to for consent from parents or guardians, legal aspects regarding adolescent sex or engagement in sex work or affordability of required routine testing for monitoring PrEP [19].

As part of the global response to reduce HIV incidence among adolescent girls and young women, the DREAMS program was introduced in 2016. Across priority districts in 15 countries, DREAMS implements a package of interventions for AGYW, their families, and male peers including partners and communities, with a central mission to "empower girls and young women and reduce HIV risk [20]. DREAMS interventions encompass a spectrum of initiatives, ranging from HIV testing, condom promotion, and social asset building to the provision of Pre-exposure Prophylaxis (PrEP), educational subsidies, business start-up kits, parenting programmes and community mobilization efforts. Social asset-building interventions are held in 'safe and private spaces' with mentors and peers for social support, curricula, and links to services. Through a comprehensive package of interventions, DREAMS seeks to address underlying and intersecting determinants of HIV vulnerability, through economic strengthening and gender equality, as a means to empower AGYW and reduce HIV risk [20].

Implementing Partners (IP) identified potential eligible AGYW for DREAMS, using 'Girl Roster' census method [21], assessing risk factors like economic status, living arrangements, marital and family situation, and educational background. IPs collaborated with community-based organizations, DREAMS mentors, and local leaders to prioritize enrolment for AGYW considered to be the most vulnerable. This included orphans, out-of-school individuals, and young mothers. This approach continues to be used for updating the programme's register and enrolment.

As part of an independent evaluation of DREAMS over six years in urban and rural settings of Kenya, we examined empowerment processes and outcomes facilitated by DREAMS interventions, i.e., whether and how DREAMS facilitated the empowerment of AGYW. These include access to resources (tangible and intangible), agency (the capacity to freely take purposeful action and pursue goals), and supportive institutional structures (as sources of both disempowerment and empowerment).

### Mixed-methods design

We used a mix-methods sequential design in which quantitative data collection and analysis preceded qualitative methods for in-depth evidence of empowerment processes and outcomes [22].

**Quantitative component.** Among representative samples of AGYW in each setting, we estimated the impact of DREAMS on structured measures of empowerment outcomes, specifically social support and generalized self-efficacy, over six years of DREAMS implementation (between 2017/18–2022). In Nairobi, 1687 AGYW aged 10–22 were enrolled into the study in 2017, and social support and self-efficacy questions were asked to those aged 15–22 (n = 889) at enrolment, over 4 rounds in 2017, 2018, 2019 and 2022. In Gem, 1023 AGYW aged 13–22 were enrolled in 2018, with social support and self-efficacy questions asked over 3 rounds, in 2018, 2019 and 2022. Characteristics of cohort participants are displayed in S1 and S2 Tables and have been described elsewhere, as have details of the cohort selection, recruitment and data collection [23].

Social support was measured in 2022 as a binary variable based on four questions relating to female networks and safe spaces (low social support was defined as a "yes" to 0–2 questions; high social support, defined as a "yes" to 3–4 questions). The four social support questions included, "Do you: "*Have a female in the community to borrow money for emergency:*"; "*Have a safe and private place to meet with female peers?*";"*Have at least one trusted female friend?*"; "*Have a female other than parent or guardian to turn to if you have a serious problem?*"

Generalized self-efficacy was measured as a binary (≥3.5 (high efficacy) vs < 3.5 (low efficacy)), combining scores from 10 questions in 2022 interviews. Examples of self-efficacy questions asked include "You can always solve difficult problems if you try hard enough"; "It is easy for you to stick to your aims and accomplish your goals"; "You are confident that you could handle unexpected events well"; and "When you are faced with a problem, you can usually find several solutions." These measures are described further in our earlier analysis of impacts by 2019 [21].

With endline data on outcomes available for 869 participants in Nairobi and 1027 in Gem, we estimated the proportions of AGYW with social support or self-efficacy in 2022 by comparing two counterfactual scenarios that all AGYW were invited to participate in DREAMS by 2022, versus no AGYW were invited into DREAMS by 2022. This matches the causal inference approach applied previously, to measure earlier impact on these same outcomes by 2019 [21]. Our primary analysis used propensity-score regression adjustment to account for the minimal set of confounding variables. We fitted a logistic regression model to predict the probability of each outcome with restriction to AGYW who were DREAMS invitees; age group and the propensity score were explanatory variables. We present these average predictions overall, and separately for younger and older AGYW.

**Qualitative methods.** In 2022, we conducted a range of in-depth and participatory activities with AGYW, implementers, and community members to investigate DREAMS' influence on pre-conditions of empowerment (access to resources, agency, and institutional structures) and achievement of empowerment outcomes. A variety of methods were used, including in-depth interviews (IDIs) and Participatory Action Learning (PAL) sessions with AGYW, Key Informant Interviews (KIIs), and Focus Group Discussions (FGDs) with adolescent boys and young men, DREAMS mentors, and community leaders. Qualitative data collection occurred between 17 October 2022 and 30 January 2023 in Gem whereas in Nairobi data collection occurred between 23 September 2022 and 30 January 2023.

## Sampling and recruitment of participants

**In-depth Interviews and Participatory Action Learning sessions with AGYW beneficiaries.** We selected AGYW participants for qualitative research from the quantitative cohorts. In *Gem*, 1027 cohort participants were interviewed in 2022 of which 592 were aged 15–24 years at that time, had previously been invited to enroll in DREAMS, and still resided in western Kenya. Of these, 177 were residing in the study area in Western Kenya. From this subgroup, individuals were selected at random to participate in qualitative research activities. A total of 59 participants consented to be part of this qualitative phase, with 23 participating in IDIs and 36 in PALs.

In *Nairobi*, 1423 cohort participants were interviewed in 2022, of which 1144 were aged 15–24 years (in 2022) and had ever been invited into DREAMS, and were traceable. From those, we invited 52 AGYW interiewees who agreed to

participate in qualitative research based on specific criteria (having ever participated in DREAMS, willing to participate in another interview, and still living in Korogocho or Viwandani); of these 52, 20 participated in IDIs and 32 in PALs.

Participants were balanced by age group, with 57 aged 15–19 years and 54 aged 20–24 years, and balanced by schooling status, though with slightly more out-of-school (59) compared to those currently enrolled (52). Most AGYW had completed Forms 3 or 4 (secondary-level education), while 12 from Nairobi and 5 from Gem had achieved university-level education. The majority of participants were not married.

*In-depth Interviews and Focus group discussions with DREAMS mentors, ABYM, CHVs and Parents*

We sought the opinions and views of other DREAMS stakeholders who were purposively sampled to participate in IDIs or FGDs. See Table 1 for summary. Data collection covered a variety of themes including empowerment of DREAMS invitees.

**Data management/processing.** Qualitative interviews were audio-recorded, transcribed, and translated into English. To enhance data quality, transcripts were reviewed, cross-checked with data collection notes and with the audio-recordings, for example to resolve unclear passages. The transcripts and data collection materials were securely stored on a password-controlled network drive with limited access, with plans to delete audio recordings after the dissemination of findings. Privacy was maintained by removing any personal or sensitive information from the transcripts.

**Data analysis.** A collaborative approach was taken to the analysis. Transcripts were divided between seven coders, who worked concurrently using a shared NVIVO 12 server database, and a shared log of coding status of transcripts. We worked in sub-teams to synthesise coded data and build themes, using a hybrid inductive-deductive approach, comparing our interpretation, for added quality assurance. We developed and entered findings into analytical data sheets in MS Word, enabling comparisons of data across cases and activity type, by setting. Reflexive, qualitative thematic analyses were conducted. Workshops and regular team meetings, at times involving the wider evaluation team, were held to discuss debriefing notes, refine codes and synthesise the coded data.

Guided by the conceptual model of women and girls' empowerment, we sought evidence of whether and how DREAMS facilitated the empowerment of AGYW participants. First we examined processes by which DREAMS enabled empowerment, specifically through influence on 'precursors' or pathways to empowerment as articulated by Kabeer (1999) [15]. Such precursors, or pre-conditions include Agency (self-determination expressed by Decision-making, Leadership and

**Table 1. Number and characteristics of participants in qualitative activities in Gem and Nairobi, 2022 (N = 234).**

| Qualitative data collection method | Participants (by gender or role and their current age in 2022) | Number of participants in each activity | |
|---|---|---|---|
| | | Gem | Nairobi |
| **In-Depth Interviews** with DREAMS participants | Adolescent girls aged 15–19 years | 12 | 10 |
| | Young women aged 20–24 years | 11 | 10 |
| **Focus Group Discussions** | Adolescent boys and young men aged 15–24years | 12 | 22 |
| | Mentors | 31 Current and former mentors, aged 20–57 years | 42 Current mentors aged 20–32 years |
| **Key Informant Interviews** | DREAMS Implementing Partner staff | 1 | 2 |
| | Parents of adolescent girls | 3 fathers and 2 mothers of DREAMS invitees | 4 mothers of DREAMS invitees |
| | Community Health Volunteers | 2 | 2 |
| **Participatory Action Learning activities** (6–10 participants per group; each group held over 3 sessions) | Adolescent girls aged 15–19 years | 20 | 16 |
| | Young women aged 20–24 years | 16 | 16 |

Collective action) and Resources (Assets, Bodily Integrity, Critical Consciousness and other resources that can be used to exercise agency). We then sought evidence of transformation through achievements, including empowerment outcomes among individual beneficiaries (voice, choice and power) and change in institutional structures (social relations, norms, laws and policies). Where relevant, we indicate findings that were either triangulated by different qualitative methods or across evaluation participant types.

## Ethical considerations

The study obtained ethical clearance from three Institutional Review Boards: KEMRI Scientific and Ethics Review Unit for Gem (KEMRI/SERU/CGHR/080/3402), AMREF Ethical and Scientific Review Committee for Nairobi (ESRC P1190 - 2022), and the London School of Hygiene & Tropical Medicine (LSHTM). A research license (NACOSTI/P/23/27412 for Nairobi) was also obtained from the National Commission for Science Technology and Innovation. Informed written consent was sought from all participants. For participants under 18 in Gem, both parental consent and the participant's assent were obtained to ensure authorization and parental awareness. In Nairobi, participants' informed consent was sought and a waiver of parental consent was granted for participants aged 15–17 years. This waiver respected the autonomy of participants when obtaining parental consent could pose risks or breaches of privacy

The research team comprised trained research assistants with experience in interviewing adolescents, including questions related to violence (emotional, physical, and sexual). Team leaders and research supervisors offered guidance, routine rest breaks, counselling and feedback sessions to support the mental well-being of researchers.

## Results

The results are presented as four dimensions of empowerment: we discuss (1) AGYW increased access to and control over tangible and intangible resources, then describe (2) enhanced agency which together resulted in (3) empowerment achievements or outcomes. We then describe (4) institutional structures that the mitigated DREAMS impact on AGYW empowerment. (Table 2).

Table 2. Themes and Sub themes related to AGYW empowerment.

| Theme | Sub themes | Impact on AGYW empowerment narratives |
|---|---|---|
| AGYW increased access to and control over tangible and intangible resources | Tangible resources Intangible resources | • DREAMS enhanced access to and control of **resources**, including highly sought-after material resources like educational subsidies and hygiene products, which reportedly reduced transactional sex through greater economic autonomy and *bodily integrity*. |
| | | • Many DREAMS participants cited greater confidence and courage (forms of critical consciousness), gained through mentor-led social asset-building curricula and social capital, enabling them to act on choices to value their sexual and reproductive health. <br> • Mentors were a highly-valued human resource, offering skills-building, counselling and training or work opportunities to AGYW. |
| Enhanced agency | Leadership Decision making Collective action (including social support) | • DREAMS participants demonstrated enhanced **agency** through *leadership* roles and *making decisions* to improve their health and future prospects, e.g., completing their education, saying no to early sex. <br> • In group activities with mentors in safe spaces, AGYW demonstrated *collective action* through female solidarity and peer support systems by pooling resources together and sharing problems with fellow women, and peers. |
| Enabling institutional structures | Social relations at peer, family, and community levels Gender norms | • DREAMS influence on supportive institutional structures was more limited. <br> • DREAMS aimed to engage the broader community, and many leaders, parents and ABYM voiced support for AGYW development and gender equality. <br> • However, contextual interventions had low reach and participants felt that the girl-centred focus of DREAMS limited its impact on prevailing patriarchal structures or widespread poverty and violence, impeding a transformative impact on young women's empowerment. |

## Acquired and strengthened resources for empowerment

This section describes the range of tangible and intangible resources that AGYW gained from DREAMS including physical assets, SRH knowledge, critical consciousness, bodily integrity and social capital that helped them act upon their choices and goals, particularly around their own SRH. ***Tangibly***, select AGYW received educational subsidies and other scholarly requirements such as uniforms, shoes, books, and solar lamps that supported their school attendance and studies. These were critical to allowing AGYW to focus on and fulfill educational aspirations.

> *Previously I was a bad girl and I knew that whenever I lacked fees I would just stay at home and sleep while waiting for mom to go look for work. But when I go to DREAMS office, I can explain and I get support – even if it's a small amount, I can go back to school…So you find that DREAMS helps one not depend on their boyfriend while focusing on your studies and future…It was like the situation was forcing me to use drugs and then you would also concentrate too much on your boyfriend. So whenever you are sent home for fees, you just go to your boyfriend. But DREAMS came and helped us a lot here in the community.*

IDI_AGYW_(20–24)_KOCH_221116_1138

> *Implementing organization helped me because when I was in primary they paid my school fees and after class eight they helped us by buying for us mattresses and sandals when I was going to high school. They also paid our fees in high school. After finishing high school they enrolled us for computer course and I got a certificate. So at least they make you busy and you cannot think about some negative things.*

PAL_(15–19)_003_VIWA_221030_0941. Participant#1.

AGYW also received menstrual hygiene materials (sanitary towels, moon cups, panties, soap) that helped to boost their sense of dignity and confidence, which also contributed to their school attendance. Accessing condoms and HIV and pregnancy tests through DREAMS also facilitated AGYW to realize SRH decisions by making such resources easily available without barriers of stigma, distance or cost.

> *The good thing is that for girls like us who come from the slums it's not always easy for us to buy pads, so the advantage of DREAMS is that you can go there anytime you need pads and get them. Sometimes we are even given soap – when you graduate after participating for a long period and undergo all those processes then you always get a certificate.*

IDI_AGYW_(20–24)_VIWA_221027_1110

Through DREAMS, some AGYW participants accessed support and inputs to start income-generating activities, such as hair dryers, nail polish, clothing stock, and sewing machines. These were complemented by acquiring financial and entrepreneurial knowledge and skills (e.g., business start-up, entrepreneurship, saving strategies).

> *There are several ways of empowering a person. You can empower them financially or with skills. So there was a time DREAMS was giving girls business startups. That helped girls be independent because you would start your own small business and earn money to support you. And also you would get the skills to run the business through the entrepreneurship training. You find that someone can save her money, feed her child and pay her rent. DREAMS also give advice about life.*

PAL 20–24 Nairobi.

AGYW acquired considerable complementary ***intangible*** resources, particularly SRH knowledge and critical consciousness, through DREAMS, according to AGYW, ABYM as well as mentors and parents. For example, learning about personal hygiene increased AGYW confidence with their menses and had knock-on effects with AGYW supporting peers

 

and siblings to navigate through menarche. Often for the first time, AGYW learned from DREAMS why, where, when, how often to be tested for HIV. Their fluency in their SRH knowledge was particularly evident when they shared advice, such as how to prevent HIV, for fictitious characters featured in vignettes discussed during the PAL sessions. Learning about HIV also motivated AGYW to know their status, even if some experienced testing as scary. They experienced HIV testing as empowering: a "negative" result motivated them to continue practicing HIV prevention and to remain negative.

As a form of 'power within', we also found evidence that AGYW learned to identify their own strengths to overcome difficulties and progress in life.

> Okay, there is a time we learnt how girls are being treated in the society. So we were told that sometimes the society feels that ladies…cannot speak out for themselves the way boys do…So when we were told that, it motivated me to advocate for myself so that I can prove to the society that girls can also stand out for themselves.
>
> IDI_AGYW_(20–24)_KOCH_221109_1022

They also gained **self-esteem** by believing in who they are and what they can do where a sense of self-worth was considered important to avoid sexual exploitation, as one participant relates:

> DREAMS has helped me handle peer pressure and I feel that if it wasn't for the implementing organization, then the peer pressure would have affected me in a very negative way. So when I joined they educated me and I boosted my self-esteem and I finished school now I am going to the next level. So I am very grateful.
>
> PAL_(15–19)_003_VIWA_221030_0941. Participant#2.

With encouragement from DREAMS mentors, AGYW learned to be **self-confident** to face difficult situations and challenges in their lives, such as job seeking, resolving family conflicts, and overcoming peer pressure. As with self-esteem, a commonly cited example was using confidence to shun unwanted advances or to insist on condom use, as an inner resource to manage gendered power imbalances.

> DREAMS told us that to have confidence so that you can prevent HIV…if you meet someone maybe he already knows he has HIV and he wants to infect you. The first thing you should do is stand up and tell him 'No' and tell him if he cannot use protection like condoms then he should just forget about it.
>
> IDI_AGYW_(15–19)_GEM_022_2022

Finally, DREAMS directly influenced AGYW **self-efficacy and self-reliance** by offering alternatives that reduced dependence on boyfriends, partners and parents so that they could progress in their lives.

> It has brought several changes but I can only talk of a few. One is that when I was still in primary, I used to depend on my mum for almost everything but now I can buy some of my dresses on own things. For example, school fees but now I can do a little job, save and pay for my next level of education. Next is that I cannot be driven by peer groups unknowingly.
>
> IDI_AGYW_(15–19)_GEM_012_2022

Quantitatively, we analysed the causal impact of DREAMS on generalized self-efficacy with the structured interview data from cohort participants in Nairobi (n = 869) and found high levels of generalized self-efficacy overall in 2022 with higher levels among DREAMS invitees than non-invitees (~71% vs 64%). The difference by DREAMS invitation was greatest

among the younger study cohort, among whom self-efficacy was ~12 percentage points higher among invitees compared to those never invited, although the statistical evidence that this difference was due to DREAMS was weak. The pattern was similar in Gem, where an increase of ~8% in self-efficacy was observed among DREAMS invitees versus non-invitees in the younger study cohort. Overall levels of self-efficacy were lower in Gem than in Nairobi, and like in Nairobi, no difference attributable to DREAMS was identified in self-efficacy among the older AGYW in Gem (Fig 1).

As a resource for women's empowerment, we examined DREAMS' role in enhancing AGYW **bodily integrity**, the realization of their rights to being healthy, secure and safe. We found AGYW gained awareness of their role in understanding, respecting, and taking care of their bodies and personal hygiene, which was complemented, for example, by receiving MH products. AGYW learned about personal safety such as their rights to bodily integrity, strategies to be safe as well as what to do in case of experiencing GBV. With gained courage from DREAMS, they spoke out and took action early, such as in cases of abusive relationships, and also they became a resource for others.

*I can say that we joined DREAMS at a young age and the first lesson we were taught…was on how one grows during puberty. What I can say is that we were taught – like when my niece starts puberty I can tell her that these things are normal so that it doesn't shatter her self-esteem because the boys will notice it and she will be trying to hide it. So, the boys will want to know what it is that she hides and that will lower her self-esteem. So, if they see that your self-esteem is high then a boy will not even touch your breast. They will know that you are too harsh and if they touch your breast*

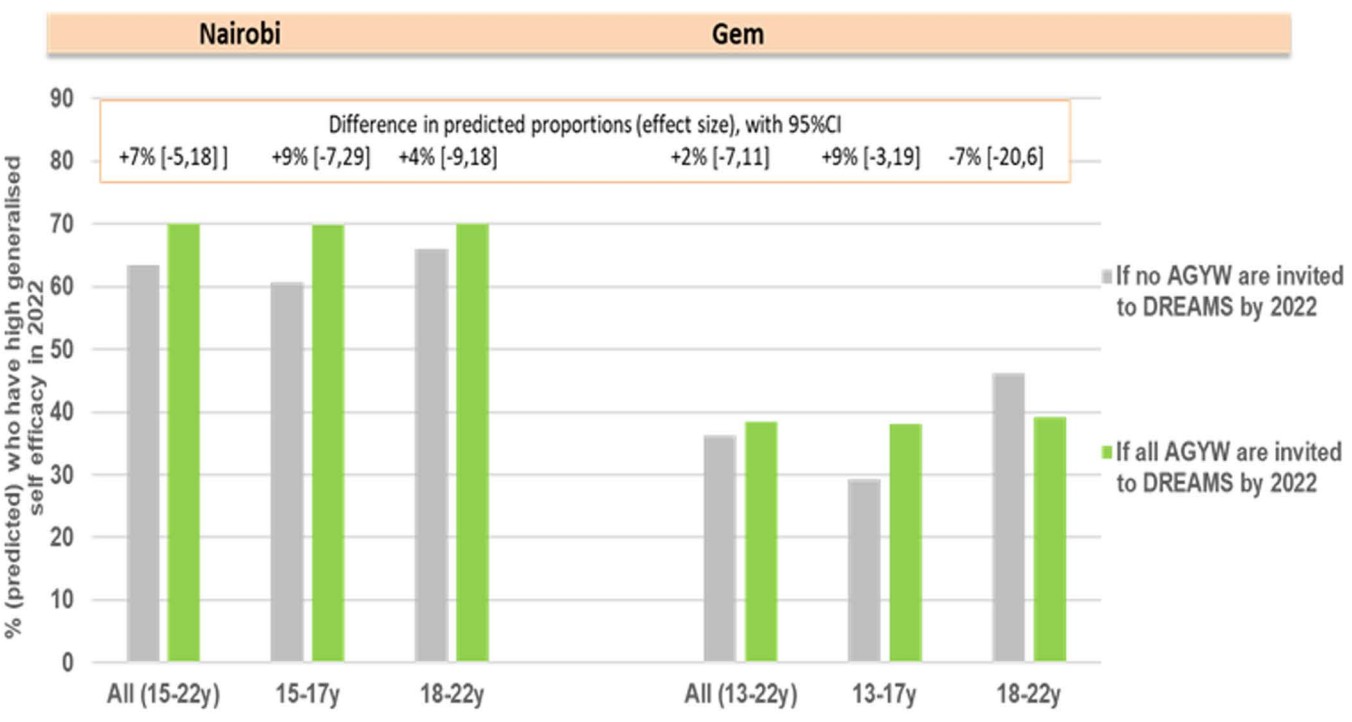

*Generalised self efficacy is a combination of scores from 10 questions, as binary (≥3.5 (high efficacy) vs <3.5 (low efficacy)).

Denominators: Nairobi (Overall n=869; 15-17 n=476; 18-22 n=393); Gem (Overall n=1027, 13-17 n=628; 18-22 n=399)

**Fig 1. Impact of DREAMS on generalized self-efficacy by 2022 in Nairobi and Gem.**

*then they can report you to the teacher…But the one who is scared and is trying to hide, the boys will start playing with her because they will know that even if they touch her then she won't do anything as she is weak.*

PAL_(20–24)_001_KOCH_221107_1009

AGYW gained ***social capital*** from building relationships and social networks with peers. This, in turn, had tangible and intangible value as sources of support and solidarity and contributed to strengthening critical consciousness, exercising leadership and participating in collective action. Social capital was particularly strengthened with mentors acting as confidants, peers, role-models, advocates as well as sources of material and emotional support. Mentors were links between AGYW and their parents as well as the IPs, passing on information, advocating on AGYW behalf and solving disputes.

*I can say that it was just the encouragement since I was in school until now. I can say that [my mentor] helped me reach where I am now. Apart from the knowledge I got from DREAMS, I also have the knowledge from the mentor because we were not always only talking about DREAMS, we would also talk about personal life.*

IDI_AGYW_(20–24)_KOCH_221117_1052

We also measured social support quantitatively with questions around female connections and access to safe and private social spaces. With structured interview data from 2022 in Nairobi (n=869), we found that social support was higher among DREAMS invitees (~57%) compared to those never invited (51%) (Fig 2). The difference by DREAMS invitation

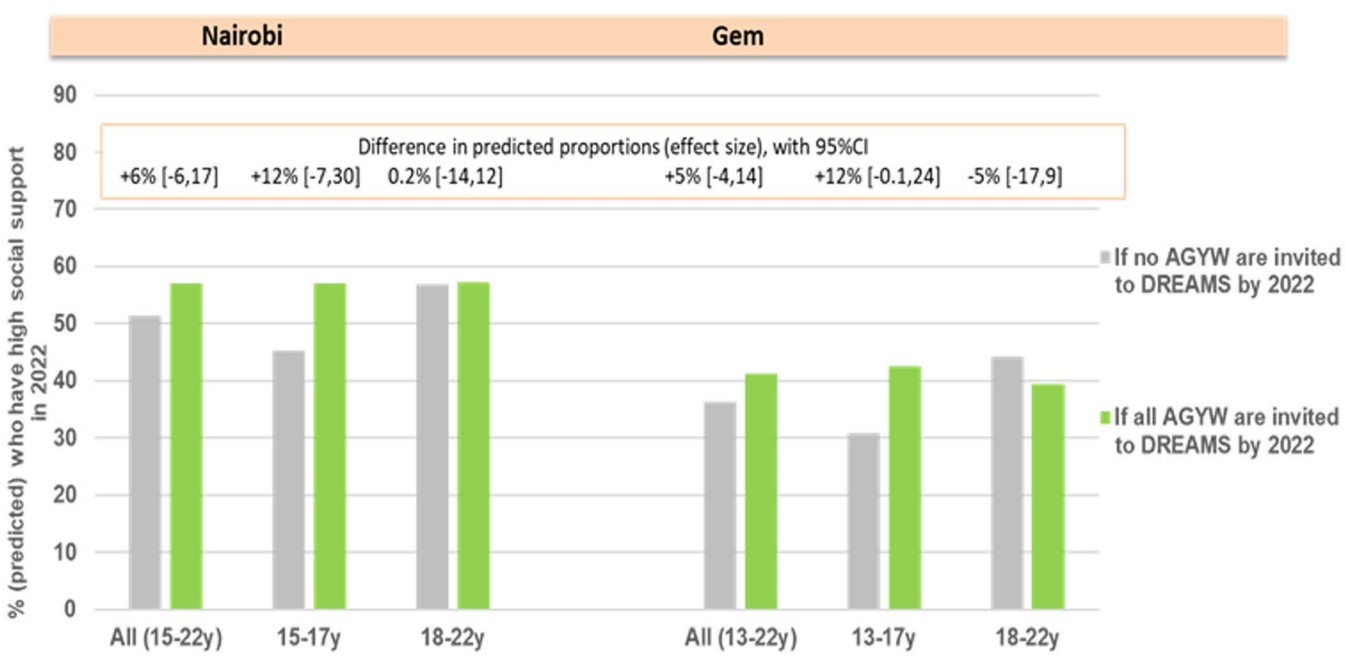

*Social support: a binary variable based on four questions relating to female networks and safe spaces (low social support, defined as a "yes" to 0-2 questions; high social support, defined as a "yes" to 3-4 questions), measured in 2022

Denominators: Nairobi (Overall n=869; 15-17 n=476; 18-22 n=393); Gem (Overall n=1027, 13-17 n=628; 18-22 n=399)

**Fig 2. Impact of DREAMS on female social support by 2022 in Nairobi and Gem.**

was greatest among the younger cohort (aged 15–17 years at cohort enrolment in 2017 and 20–22 years at end-line in 2022), among whom social support was 10–12 percentage points higher for DREAMS invitees compared to non-invitees (~56% vs 45%, respectively). However, the statistical evidence that these differences were due to DREAMS was weak (namely because most AGYW were invited to DREAMS and the comparison group of non-invitees was small).

We observed a similar pattern in Gem (n = 1027), with social support higher for DREAMS invitees (~43%) compared to those not invited by 2022 (31%), among the younger cohort (aged 13–17 at study enrolment in 2018). No evidence was found for an impact of DREAMS on social support among the older study cohort in Gem (aged 18–22 at enrolment in 2018). Compared to Nairobi, the level of social support among DREAMS and non-DREAMS participants appeared lower in Gem.

**Agency enhanced**

As a second dimension of empowerment, we found evidence of enhanced AGYW capacity, due to DREAMS, to ***explore their priorities, set goals, and act on important life decisions*** concerning education, early sex, and intimate relationships such a marriage. With such decisions, AGYW prioritized personal goals and resisted negative peer influences about, for example, drug use and male sexual advances.

*Because like right now I can make my own choices. You cannot find that a man just treats you the way he wants. You know that as a girl you have big dreams to achieve and you work towards achieving them first, instead of doing what the man wants. Maybe if you have a boyfriend, you must not follow them. You have your own goals in life so you move on with it.*

IDI_AGYW_(20–24)_VIWA_281022_0937

For sexually active AGYW, they acted upon information they received through DREAMS to minimize risk of HIV and pregnancy by using condoms and other contraception, testing for HIV, knowing their partner's status and/or taking PrEP.

*We had a topic in DREAMS called healthy choices. We were taught things like how one should prevent themselves from HIV/AIDS and STIs. So I feel empowered because I haven't had the disease. So maybe using the preventions and precautions that we learned there has helped me to avoid the diseases.*

IDI_AGYW_(20–24)_VIWA_031122_0949

Agency is also expressed through ***collective action***, in part as an outcome of acquired social capital, and we found strong evidence of DREAMS fostering solidarity, including the mutual support of peers and mentors, from which AGYW drew strength. This support came in the form of moral and material assistance with participants reported sharing their problems and pooling their ideas as well as their resources (e.g., saving money together through table-banking in 'chamas').

*When people were called to go, I felt I would rather go and learn together with my peers instead of just sitting at home. While you are there, when you are sharing with many other people you will learn from others because people think differently, what I am thinking is different from what another person thinks. So we would share ideas.*

IDI_AGYW_(20–24)_GEM_008_2022

The ideas, confidence, and shared identity that many gained from DREAMS, a form of 'power with', helped AGYW to exercise agency around their sexual health. For example, some were motivated to get HIV testing when this was organized together in groups.

Participants also exercised agency to help others, applying the benefits of DREAMS to support their siblings, parents, and their own children's health. For example, participants were able to share lessons from DREAMS with close peers, expanding the influence of DREAMS through social networks.

*Right now I have a sister in class seven and during her first period I can show her how to wear the pads because she cannot be taught by the mother. During my first period I felt embarrassed at school. So I took my skirt – it was only one day – so when I saw someone coming I would hide it. When my mother found it she asked what it was. She insulted me. But since I went to DREAMS, my sister started having her periods and I told her that it was normal and she was growing.*

PAL 20-24y Viwandani.

DREAMS participants also shared knowledge and skills they acquired from DREAMS with non-DREAMS peers, for example, by encouraging others to shun peer pressure and to avoid decisions that may compromise their future.

*I always use what my mentor taught me to convince other girls not to enter peer pressure. I am telling them not to fall for traps set by men because after getting pregnant they will be dumped and they will end up dropping out of school while the boys continue with their education.*

IDI_AGYW_(15–19)_GEM_016_2022

Gaining social capital through such solidarity was valued, for example, as helping others was considered *a good thing because you do not know who will come to your aid tomorrow or who will help your child in future* [FGD Mentors, Gem].

**Empowerment outcomes to prevent HIV and pursue SRH: choice and opportunities, voice and financial autonomy**

We found ample evidence that DREAMS **expanded the repertoire of choices** for AGYW to protect themselves from HIV and prevent pregnancy, which resulted in part from the provision of assets described above. These allowed for greater options for exercising of agency, such as pursuing an education, and greater financial autonomy in the pursuit of goals by decreasing AGYW dependence, particularly on men, and avoiding possible exploitation.

*I see myself empowered in that if it were not about DREAMS that taught about health education maybe I could have joined the girls...Some of them didn't finish their schooling and they dropped out. So, I see myself empowered in that I managed to finish my form four…I too could have dropped out of school...They taught us that some of the girls are not able to get the sanitary towels and for that case they're cheated by men that would use them and then give them the sanitary towels. But by giving me those sanitary towels, I feel empowered.*

IDI_AGYW_(15_19)_GEM_013

Expansion of choice entailed DREAMS **increasing the knowledge of choices**, which supported AGYW making informed choices, particularly critical life choice such as when to have sex, with whom to have sex and under what conditions. As part of this, AGYW learned of their rights to make their own choice.

*So, it depends, let's say your mother will see that you are now of age and a grown woman, and so they want a way of how you can leave the home...She will tell you 'I think now you are old enough to have a boyfriend, go to your boyfriend to buy you even sugar you come with,' such that if you do not bring the sugar it is trouble and this forces you to do what the parent wants. But if you were going to DREAMS, you have the rights to refuse and tell her…So if you are this side, you can help yourself better….*

IDI_AGYW_(20_24)_GEM_004_2022

Along with more choices and increased knowledge of these, DREAMS enhanced AGYW *capacity to make choices in the interests of their own health*, such as reducing SRH risks through abstinence, delaying marriage, and having courage to make independent choices and withstanding peer pressure.

> I:What do you think about DREAMS as a program?.
>
> R: …it has helped girls stand for themselves like they have the ability to make correct choices, either when there is danger or somethings just like you want to decide if you want to do this or not on anything, so you choose whether you want to protect yourself. For example if you are in a relationship with someone you decide to stand for it, like you have to protect yourself and know the health status of each of you.
>
> IDI_AGYW_(20–24)_GEM_001_2022

The expansion of choice and enhanced capacity to make independent choices had a longer-term impact when these challenged social inequalities, or when DREAMS-provided opportunities resulted in further opportunity and choice. DREAMS expansion of choices concerned not only providing alternatives and options but also *expansion of the realm of the possible* for DREAMS participants.

> I:Was there a time these decisions looked impossible but it got to a time when they were possible?.
>
> R:Yes…like in the slums, the mentality of people is that after completing class eight or form four that's the limit. All they think about is when you finish form four you get married. So if you are married after form four in the slums then it's like you've gained something. But for me, I felt like one must not be married. So before I used to feel that there was no hope in the community but after joining DREAMS, they motivate you and you get that self-drive so you think beyond.
>
> IDI_AGYW_(20–24)_VIWA_221026_1104

DREAMS participants as well as other evaluation informants mentioned several ways in which the programme led to *enhanced AGYW voice*: speaking up with the newly-gained confidence, articulating their concerns and interests, being heard and shaping and sharing in discussions and decisions that concern their lives and lives of others. Enhanced voice was a result, in part, from AGYW gaining critical consciousness as described above that DREAMs contributed to. Examples of enhanced voice included AGYW better managing sexual relationships by saying no confidently and meaning it (exercising their right to refuse sex) and using non-verbal communication such as direct eye contact, as described by a number of participants. This led to AGYW being heard and respected, such as avoiding unwanted sexual encounters or acting against abusive relationships.

In addition, AGYW developed capacity to have better communication with partners and parents because of having enhanced voice with some parents acknowledging changes in AGYW in their ability to communicate about their health, which was not possible before engagement in DREAMS. This enhanced voice also led to better relationships with partners who supported AGYW to continue participation in DREAMS.

AGYW expansion of choice and opportunities was also enhanced with *increased financial autonomy* and strengthened financial authority to make own decisions and control over finances. For example, AGYW's strengthened financial management knowledge and skills improved their practice for financial autonomy such as from engaging in different saving schemes and in income generating activities. These increased their financial options by having their own money or increased access to financial support that allowed them to be less reliant on others. There was evidence on avoidance of transactional sexual relationships resulting from greater financial autonomy as well as avoidance of peer pressure to look for boyfriends to meet material needs.

R: I had not been taught about the things I had told you earlier... like I have told you, I would have succumbed to peer pressure. Like I would have seen a lady friend of mine with a boyfriend who gives her money and also do the same. So that was the difference.

I: Does that also have a relationship with being empowered?

R: Yeah there is, because there after you had been taught how you can open a bank account, you can establish your home bank, you can save in a bank, you can save in a SACCO, etc. Then if you know you have savings, and somebody says to you 'I want to give you this much money so that I can do this with you'. You have a right to tell him no, I can't do that.

(IDI_AGYW_(15_19)_GEM_020_2022)

Despite the efforts to achieve economic outcomes for the AGYW, there were challenges experienced regarding establishing and maintaining businesses or utilising developed vocational skills, based on observations from parents and mentors.

## Persistent institutional structures

As the fourth dimension of empowerment, the evaluation sought evidence of DREAMS impact on wider social structures, understood as institutional structures (BMGF, 2017), as an important precursor for AGYW empowerment and found limited effect. AGYW did describe gender norms associated with their empowerment, which they in part attributed to DREAMS. For example, DREAMS challenged the common notion that sexual violence is a private family affair with DREAMS mentors and IPs sometimes intervening to support AGYW experiencing violence by partners or relatives.

However, the evaluation identified ways that dominant gender inequitable norms posed challenges for AGYW, and remained unchanged. For example, AGYW were more stigmatized when seeking sexual and reproductive health services such as contraceptives, condoms and PrEP compared to men seeking similar services. Religious gender norms restricted AGYW access to FP and other SRH information. In other examples, we found DREAMS facilitators projecting societal expectations of young women, for example, such as practicing abstinence, wearing modest clothing or how women should be obedient to men, as a form of respect, thereby possibly reproducing dominant gender inequitable norms. "*With DREAMS, girls are taught to be obedient*" (IDI_AGYW_15_19_GEM_022_2022). We also uncovered male-dominant gender norms, exclusively from ABYM informants, such as projections of AGYW as vulnerable and weak given their gender and men's role as breadwinners and being physically stronger.

*So the reason why DREAMS, as an organization, was concentrating on girls, is because girls are termed as vulnerable in the society. They are prone to challenges. For example, when conflicts occur, you realize that a bigger percentage of those affected are the girls. They are vulnerable. For example, there are some tasks or jobs that girls cannot do, but men can hustle and do. And this one leads the girls into engaging in certain activities that make them vulnerable. For example, construction of houses. Girls rarely engage in such works, yet they are the most available jobs in the country. So, this will make them engage in some ways of looking for money which are not appropriate like prostitution, which can later bring them problems in their lives.*

FGD_ABYM_GEM_001_2022

More generally concerning ABYM, we found limited involvement with DREAMS, typically confined to some SRH classes and offers of voluntary medical male circumcision for some male community members. Still, ABYM directly and indirectly benefitted such as learning, sometimes alongside AGYW, about HIV prevention and family planning, resulting by their accounts, in practicing safe sex. Indirectly and according to some informants, ABYM may also have benefitted with

household finances being freed up when their sisters' school fees were covered by DREAMS, which were then used for sons.

Overall, however, ABYM reported feeling sidelined, abandoned and discriminated against by the programme. They uniformly expressed being un-recognized and misunderstood such as assumed to being lazy, problematic and powerful. They indicated they too have needs, and they too are vulnerable and face discrimination, such as from the police.

*And also they should also create time to listen to us as boys so that they don't say that boys are strong. So if they are able to listen to us they will know that we also do go through a lot.*

FGD_ABYM_KOCH_221003_1003

AGYW also shared concerns that substantive exclusion of ABYM will create greater inequity and discord among women and men in the future: future husbands will feel threatened and not support their empowered wives, which ultimately constrains efforts to empower AGYW.

*So when they consider the girls, they should also consider the boys, because the moment I am empowered and the boy is not empowered, I will grow and all that but in the end I'll be with a man and maybe somehow the men will just be from the community. So when it comes to home, you find that there are conflicts at home. So that's why you find that most girls are single ladies or boos-ladies because you find that they feel that they are too superior to marry a useless person. She will call the man a useless person because he is not empowered.*

IDI_AGYW_(20–24)_KOCH_221123_1147

Within households, there were reports of less conflict between parents and girls, from teachings through the Family Matters course for parents, focused on understanding AGYW, as well on inter-personal communications that emphasized AGYW respect for and obedience to parents. Mentors also reported less conflict among couples that include DREAMS participants, in part due to couple counseling and family planning provided by the programme. At the same time, some AGYW reported backlash from husbands in response to their participation in and benefit from DREAMS, in part from misunderstanding what DREAMS is about, as well as resentment about AGYW progressing – economically and otherwise – due to DREAMS.

At the community level, the evaluation found evidence of limited impact despite DREAMS collaborating with existing community services such as rescue centers and police toll-free numbers in cases of GBV. AGYW were made aware of their rights but did not always seek support and also realized that their rights would not always respected. For example, reporting cases of rape was delayed or resolved informally without consequence for the perpetrator. In other cases, AGYW were unable to access community resources such as Village and Saving Loan Associations because they were not of legal age to do so.

## Discussion

Part of the rationale for this 2022 evaluation of DREAMS was to assess the programme's influence on the empowerment of AGYW to prevent HIV by allowing for sufficient time to observe longer-term effects (over 6 years) [24]. Overall we found evidence of AGYW accessing and utilizing resources realized from the DREAMS programme, particularly intangible resources such as 'power within' which in turn facilitated their agency and 'power-to' resulting in empowerment achievements, particularly, the expansion of possibilities and informed choices that they acted on [25]. The realization of intangible resources distinguishes DREAMS from SRH programmes focused on service delivery: resources and agency are needed not only to utilize such SRH services effectively, but also to sustain AGYW efficacy beyond the programme [26]. Also, findings from this study illustrate that strengthened AGYW agency extended beyond DREAMS participants by

becoming a basis for leadership of others. DREAMS facilitated collective action that enhanced individual "power-to" by strengthening"power-with" other young women [27].

Such expansion of empowerment can also sustain DREAMS impacts. AGYW's horizons of "what is possible" through the expansion of life choices and possibilities provided the basis for alternative futures to what they saw for themselves and in their peers. This was experienced as an expansion of opportunities that, together with enhanced resources, increased capacities to act to prevent HIV. Related studies have shown that multipronged interventions that integrate social, reproductive health, and economic assets building can have a greater impact on AGYW SRH outcomes compared to individual component interventions, for better reproductive health outcomes [28,29]).

Our findings also reiterate the understanding that empowerment is not a singular process: Cornwall et al., (2014), refer to the plurality of possible pathways of empowerment [30]. Which path an AGYW takes is context specific, in part due to the specific social institutions that are at the foundations of a particular AGYW experience of disempowerment.

Also, empowerment is not a linear process but an iterative one where resistance, if not backlash, can be encountered either from husbands/partners, family or community members [31]. While AGYW experienced shifts in beliefs in themselves and sometimes their parents' beliefs about AGYW, dominant gender norms proved to be always present in various and complex ways, such as mentors sometimes reproducing norms antithetical to AGYW empowerment [32]. Also, ABYM, while overall understanding of the need for and supportive of AGYW empowerment, at times also maintained dominant gender views that reinforced tropes of vulnerable women and strong men by nature of their genders. Their endorsement of their female counterparts seemed more about empathy for the opportunities for material and economic advancement that DREAMS was perceived to provide, which is understandable given the low-resources conditions of DREAMS communities.

Both these examples of mentors and ABYM underscore the importance of understanding the empowerment of AGYW within the social relations in which they experience disempowerment. While the DREAMS programme engaged with ABYM and household and community members, it was not through the social relations of gender that AGYW experience constraining gender norms within their households, communities and beyond. In other words, DREAMS almost exclusive focus on AGYW meant alienating them (and others) from their actual contexts and social relations. Revisiting a gender synchronized approach [33], which recognizes and works with both men and women as conduits for gender norms, may be warranted, particularly to work at the level relevant institutional structures [34]. Such an approach calls for context-specific design of working with all genders while also maintaining a focus on AGYW empowerment. In a specific example, we found evidence that DREAMS encouraged young women to exercise their right to refuse sex, but no evidence that males were sensitized about sexual and reproductive rights (women's or their own) including consent. or freedom from sexual coercion.

Future considerations of DREAMS may also benefit from the analytical and programming lens offered by gender transformative approaches (GTA) to health promotion and AGYW empowerment, beyond DREAMS current multi-level intervention approach. Such an approach addresses the foundations of gender inequity and recognizes and builds upon the strengths of AGYW rather than their deficits [35]. The experience of evaluation participants with participatory action learning methods is an example of this, as many felt empowered when asked for their advice or views on scenarios and vignettes.

GTA's also require working through and with social institutions, as they are at the foundations of inequitable social relations. For MacArthur et al. (2022), a focus on systems is particularly enlightening considering the evaluation findings [36]. This not only requires an emphasis on norms (both restrictive but also liberating) and their reproduction (such as in the case of mentors also being conduits of lived norms) but also on working with and through households (families) and communities.

For example, while the evaluation focused on DREAMS interventions, AGYW also identified a range other influences on their knowledge, beliefs and actions, most of which are from their immediate surroundings, including peers, family

members such as parents and community-based organizations. Working with these extant sources of empowerment presumably not only makes them appropriate for and acceptable by AGYW, but also renders efforts to empower AGYW more sustainable. They can also provide a basis for inter-generational impact in that next generations of both AGYW and ABYM can also benefit from AGYW empowerment as "future claims and expectations" [15].

Our findings can be applicable to other contexts across the 15 countries where a comparable core package of DREAMS interventions is being implemented [37]. In such settings, a girl-centered approach will face similar limitations until responsibility for HIV and gender outcomes are shared across gatekeepers and institutions. As other programmes adopt multi-component or gender-based approaches to HIV prevention, these findings highlight the importance of anticipating and examining social and institutional hurdles before implementation.

## Strengths and limitations

Key strengths of our evaluation include the scale and representativeness of quantitative data (enabling causal analysis over six years of DREAMS implementation) combined with rich qualitative data for deeper insights and diverse perspectives. The evaluation was conducted over six years in established research settings by experienced teams. The collaborative nature of the analysis, by a multi-disciplinary team including those who coordinated the data collection, enhanced the quality and validity of the findings.

Limitations nonetheless included social desirability bias, particularly as some participants, e.g., AGYW, mentors and IPs may have been motivated in their contributions by either fearing compromising their benefits from the DREAMS programme or that they may personally gain by providing particular answers. However, many spoke openly about their challenges and criticisms, so the extent of this bias may be limited. Also, we had a limited number of measures to quantify constructs within Kabeer's model for women's empowerment and we encourage the development of measures to quantify 'achievement' of empowerment outcomes. These could be aligned with age- and gender-specific measures of gender-transformative outcomes. Finally, few AGYW in the study cohorts had *not* been invited to DREAMS by 2022 (given wide reach in these settings), limiting participant numbers to identify an impact with greater statistical certainty.

## Conclusion

We found compelling evidence that DREAMS strengthened many young women's agency – both intrinsic and collective – facilitated by their access to and control over resources to help reduce HIV risk. Quantitative evidence showed stronger DREAMS effects for social support and self-efficacy in younger participants, demonstrating the value of reaching adolescents for interventions at an early age. The strengthened agency and resources we observed are important precursors to empowerment outcomes but insufficient for the transformation of power relations without sufficient supportive institutional structures including gender-equitable norms and viable livelihood opportunities. Expanding DREAMS' impact on young women's empowerment with complementary policy, market, and media strategies could help to address the social and economic foundations of young women's heightened HIV risk.

## Supporting information

**S1 Table. Description of the characteristics of the AGYW at enrolment in Gem 2018, by age and invitation to DREAMS.**
(XLSX)

**S2 Table. Description of the characteristics of the AGYW at enrolment in Nairobi 2017, by age and invitation to DREAMS.**
(XLSX)

**S1 Checklist. Human participants research checklist.**
(DOCX)

## Acknowledgments

We would like to acknowledge the participants in Gem, Korogocho and Viwandani for their time, information, and support. We are also grateful for the efforts of the project field team, and the project and data management staff who helped to ensure high quality data.

## Author contributions

**Conceptualization:** Elizabeth Kemigisha, Venetia Baker, Annabelle Gourlay, Stephen Gakuo, Sarah Mulwa, Sammy Khagayi, Daniel Kwaro, Sian Floyd, Abdhalah Ziraba, Isolde Birdthistle.

**Data curation:** Elizabeth Kemigisha, Jane Osindo, Annabelle Gourlay, Sarah Mulwa, Faith Magut, Thomas Gachie, Moses Otieno, Sian Floyd, Isolde Birdthistle.

**Formal analysis:** Elizabeth Kemigisha, Jane Osindo, Vivienne Kamire, Venetia Baker, Annabelle Gourlay, Sarah Mulwa, Faith Magut, Thomas Gachie, Moses Otieno, Sammy Khagayi, Sian Floyd, Isolde Birdthistle.

**Funding acquisition:** Daniel Kwaro, Sian Floyd, Abdhalah Ziraba, Isolde Birdthistle.

**Methodology:** Elizabeth Kemigisha, Venetia Baker, Annabelle Gourlay, Stephen Gakuo, Sarah Mulwa, Faith Magut, Thomas Gachie, Sammy Khagayi, Daniel Kwaro, Sian Floyd, Isolde Birdthistle.

**Project administration:** Jane Osindo, Vivienne Kamire, Sian Floyd, Abdhalah Ziraba, Isolde Birdthistle.

**Supervision:** Jane Osindo, Vivienne Kamire, Annabelle Gourlay, Sian Floyd, Abdhalah Ziraba, Isolde Birdthistle.

**Validation:** Elizabeth Kemigisha, Vivienne Kamire, Venetia Baker, Annabelle Gourlay, Stephen Gakuo, Sarah Mulwa, Faith Magut, Thomas Gachie, Moses Otieno, Daniel Kwaro, Abdhalah Ziraba.

**Visualization:** Sarah Mulwa, Faith Magut, Moses Otieno, Isolde Birdthistle.

**Writing – original draft:** Elizabeth Kemigisha, Isolde Birdthistle.

**Writing – review & editing:** Elizabeth Kemigisha, Jane Osindo, Vivienne Kamire, Venetia Baker, Annabelle Gourlay, Stephen Gakuo, Sarah Mulwa, Faith Magut, Thomas Gachie, Moses Otieno, Sammy Khagayi, Daniel Kwaro, Sian Floyd, Abdhalah Ziraba, Isolde Birdthistle.

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
