## [Decision Letter · Decision Letter 0]

PGPH-D-24-02060

Impact of the DREAMS Partnership on adolescent girls and young women’s empowerment for HIV risk reduction: a mixed methods evaluation in Kenya

Dear Dr. Kemigisha,

Thank you for submitting your manuscript to PLOS Global Public Health. After careful consideration, we feel that it has merit but does not fully meet PLOS Global Public Health’s publication criteria as it currently stands. Therefore, we invite you to submit a revised version of the manuscript that addresses the points raised during the review process.

We look forward to receiving your revised manuscript.

Kind regards,

Nancy Angeline Gnanaselvam

Academic Editor

Journal Requirements:

 1. Please include a complete copy of PLOS’ questionnaire on inclusivity in global research in your revised manuscript. Our policy for research in this area aims to improve transparency in the reporting of research performed outside of researchers’ own country or community. The policy applies to researchers who have travelled to a different country to conduct research, research with Indigenous populations or their lands, and research on cultural artefacts. The questionnaire can also be requested at the journal’s discretion for any other submissions, even if these conditions are not met.  Please find more information on the policy and a link to download a blank copy of the questionnaire here: https://journals.plos.org/globalpublichealth/s/best-practices-in-research-reporting. Please upload a completed version of your questionnaire as Supporting Information when you resubmit your manuscript. 2. Your current Financial Disclosure states, “The funders had no role in study design, data collection and analysis, decision to publish, or preparation of the manuscript.”. However, your funding information on the submission form indicates that you received funding from “Bill and Melinda Gates Foundation INV-027727 Abdhalah Ziraba”. Please indicate by return email the full and correct funding information for your study and confirm the order in which funding contributions should appear. Please be sure to indicate whether the funders played any role in the study design, data collection and analysis, decision to publish, or preparation of the manuscript. 3. In the online submission form, you indicated that "Data sources will be available upon request from the authors.".  All PLOS journals now require all data underlying the findings described in their manuscript to be freely available to other researchers, either 1. In a public repository, 2. Within the manuscript itself, or 3. Uploaded as supplementary information. This policy applies to all data except where public deposition would breach compliance with the protocol approved by your research ethics board. If your data cannot be made publicly available for ethical or legal reasons (e.g., public availability would compromise patient privacy), please explain your reasons by return email and your exemption request will be escalated to the editor for approval. Your exemption request will be handled independently and will not hold up the peer review process, but will need to be resolved should your manuscript be accepted for publication. One of the Editorial team will then be in touch if there are any issues. 4. Figure 1: Please confirm whether you drew the images / clip-art within the figure panels by hand. If you did not draw the images, please provide (a) a link to the source of the images or icons and their license / terms of use; or (b) written permission from the copyright holder to publish the images or icons under our CC-BY 4.0 license. Alternatively, you may replace the images with open source alternatives. See these open source resources you may use to replace images / clip-art:- https://commons.wikimedia.org-
https://openclipart.org/ 5. We have noticed that you have a list of Supporting Information legends in your manuscript. However, the uploaded files are figures which already uploaded in the file inventory. Kindly remove the citation of "Supporting Information".

Additional Editor Comments (if provided):

While the study intervention to address HIV among younger population is commendable, the entire paper reads as a feedback to the DREAMS project

1. Authors need to describe the scalability and sustainability of this intervention and other similar interventions to empower young population infected with HIV in resource poor and conflict settings

2. Themes and sub themes are insufficiently described

3. Enthusiastic consent and sexual coercion is simplified in the study results description. Please describe in detail and explain the challenges in teaching such concepts to young population in the discussion

4. In Nairobi did any any child refuse to participate in the study? Of those who participated without parental consent, what were challenges. Can these children be termed as emancipated due to abuse or neglect or absent parents? What additional protection measures were taken when assent was obtained from these children to ensure autonomy was not compromised.

5. Clarify the need for Figure 1 when BMGF Document is already cited in the article

6. Child abuse incidents are being described in this study. What are the preventive measures taken by investigators to address vicarious trauma among local investigators?

7. Please provide the CRediT statement for the authors

8. Statistical analysis needs further description in methods and clarify whether it is a mixed methods study design

Reviewers' comments:

Reviewer's Responses to Questions

**Comments to the Author**

1. Does this manuscript meet PLOS Global Public Health’s publication criteria ? Is the manuscript technically sound, and do the data support the conclusions? The manuscript must describe methodologically and ethically rigorous research with conclusions that are appropriately drawn based on the data presented.

Reviewer #1: Yes

Reviewer #2: Yes

2. Has the statistical analysis been performed appropriately and rigorously?

Reviewer #1: Yes

Reviewer #2: I don't know

3. Have the authors made all data underlying the findings in their manuscript fully available (please refer to the Data Availability Statement at the start of the manuscript PDF file)?

Reviewer #1: No

Reviewer #2: Yes

4. Is the manuscript presented in an intelligible fashion and written in standard English?

Reviewer #1: Yes

Reviewer #2: Yes

5. Review Comments to the Author

Reviewer #1: 1. Baseline data in Table 1 is not clear, what is the unit? Is it frequency, percentage?

I believe it is n, but it will be great if it is mentioned.

Also, the total number of samples should be mentioned on top of the table.

2. The quantitative aspect of the study can be best explained in a tabular form.

Reviewer #2: Abstract

The abstract states the study's elements—introduction, methods, results, and conclusions—facilitating a coherent progression for readers.

You can consider explicitly writing the research questions or objectives, such as “This study aims to evaluate the extent to which DREAMS interventions have promoted the empowerment of adolescent girls and young women (AGYW) in Kenya”. Consider including it in abstract as well as in the introduction.

The statement "we discovered insufficient evidence of a causal effect" (line 36) can be further clarified. What is the meaning of "weak evidence" in this context? Does it pertain to statistically negligible outcomes, minimal effect sizes, or other constraints?

Authors may want to mention any prominent themes or patterns that arose from the interviews and conversations, beyond specific examples such as "pooling resources together" (line 42).

The abstract references empowerment precursors such as "resources, agency, and institutional structures" (line 29), which are essential notions in empowerment theory. It may be beneficial to reference the theoretical framework underpinning this study, whether it is the specific DREAMS framework or another recognised empowerment paradigm.

Introduction

It would be beneficial to offer a concise explanation or operational definition of "gender-transformative". How does the study conceptualise "gender-transformative" in relation to DREAMS? (line 83)

The word "structural determinants" (line 69) might benefit from a concise explanation or an illustrative example for people unfamiliar with the idea.

Methods

The methods section is detailed and robust, but with clearer descriptions of the quantitative analysis, more integration of the conceptual framework, it would better demonstrate the rigor of the research.

The section on how participants were selected—both for the overall cohort and for the qualitative components—demonstrates methodological rigor.

The operationalisation of social support and self-efficacy is clear, but it would be helpful to have a short description of the specific questions that were used to measure them.

The statistical methods used for causal analysis could be explained in more depth. How were the different possible outcomes (early invitees vs. later invitees) analysed?

The conceptual model of women and girls’ empowerment, based on Kabeer’s framework, is mentioned, but not fully included in the description of the methods. It would benefit readers to show how the methods were chosen, interview guides were made, and the coding process was carried out in qualitative analysis.

The part on results gives a complete and well-organised picture of how DREAMS has affected the empowerment of AGYW. The section can be made even stronger by showing how outcomes vary by the age group (Adolescents & young women), making the statistical analysis clearer and describing more about the institutional hurdles.

Discussion

The discussion section presents a thorough interpretation of the findings. The conversation will provide even more insightful information for upcoming programmes if it places more emphasis on quantifiable outcomes, examines institutional hurdles in greater detail, and makes age specific recommendations.

6. PLOS authors have the option to publish the peer review history of their article (what does this mean? ). If published, this will include your full peer review and any attached files.

**Do you want your identity to be public for this peer review?** For information about this choice, including consent withdrawal, please see our Privacy Policy .

Reviewer #1: No

Reviewer #2: No

---

## [Decision Letter · Decision Letter 1]

PGPH-D-24-02060R1

Impact of the DREAMS Partnership on adolescent girls and young women’s empowerment for HIV risk reduction: a mixed methods evaluation in Kenya

Dear Dr. Kemigisha,

Thank you for submitting your manuscript to PLOS Global Public Health. After careful consideration, we feel that it has merit but does not fully meet PLOS Global Public Health’s publication criteria as it currently stands. Therefore, we invite you to submit a revised version of the manuscript that addresses the points raised during the review process.

We look forward to receiving your revised manuscript.

Kind regards,

Nancy Angeline Gnanaselvam

Academic Editor

Journal Requirements:

Additional Editor Comments (if provided):

1. Introduction: A prominent example of multi-level programming to prevent HIV risk among adolescent

85 girls and young women is the DREAMS Partnership (Determined, Resilient, Empowered, AIDS-free,

86 Mentored, and Safe) led by PEPFAR and private sector partners.

These details especially the funder details can be put in reference and not in manuscript.

2. A conceptual framework would be much helpful for readers to understand the themes visually

Reviewers' comments:

Reviewer's Responses to Questions

**Comments to the Author**

1. If the authors have adequately addressed your comments raised in a previous round of review and you feel that this manuscript is now acceptable for publication, you may indicate that here to bypass the “Comments to the Author” section, enter your conflict of interest statement in the “Confidential to Editor” section, and submit your "Accept" recommendation.

Reviewer #1: (No Response)

2. Does this manuscript meet PLOS Global Public Health’s publication criteria ? Is the manuscript technically sound, and do the data support the conclusions? The manuscript must describe methodologically and ethically rigorous research with conclusions that are appropriately drawn based on the data presented.

Reviewer #1: Yes

3. Has the statistical analysis been performed appropriately and rigorously?

Reviewer #1: I don't know

4. Have the authors made all data underlying the findings in their manuscript fully available (please refer to the Data Availability Statement at the start of the manuscript PDF file)?

Reviewer #1: Yes

5. Is the manuscript presented in an intelligible fashion and written in standard English?

Reviewer #1: Yes

6. Review Comments to the Author

Reviewer #1: Comments have not been addressed in the main paper.

7. PLOS authors have the option to publish the peer review history of their article (what does this mean? ). If published, this will include your full peer review and any attached files.

**Do you want your identity to be public for this peer review?** For information about this choice, including consent withdrawal, please see our Privacy Policy .

Reviewer #1: No

---

## [Editor Report · Decision Letter 2]

PGPH-D-24-02060R2

Impact of the DREAMS Partnership on adolescent girls and young women’s empowerment for HIV risk reduction: a mixed methods evaluation in Kenya

Dear Dr. Kemigisha,

Thank you for submitting your manuscript to PLOS Global Public Health. After careful consideration, we feel that it has merit but does not fully meet PLOS Global Public Health’s publication criteria as it currently stands. Therefore, we invite you to submit a revised version of the manuscript that addresses the points raised during the review process.

We look forward to receiving your revised manuscript.

Kind regards,

Nancy Angeline Gnanaselvam

Academic Editor

Journal Requirements:

Additional Editor Comments (if provided):

Thank you for your prompt and adequate response to the queries. The paper looks brilliant.

I kindly request you to elaborate further in the "Evaluation research contexts: DREAMS in Kenya" section by shifting the DREAMS description in the introduction section to methods section

In the introduction please stick to the following sub sections

HIV and social determinants of health

Psychological aspects of HIV

Rights, dignity and legal aspects and any other aspects relevant to the study

There is a need to add a study settings section in methods in which you elaborate the prevalence of HIV in relevant populations in the study setting, any on going geopolitical issues, conflicts in the study area, legal issues, health care access available in the study area (with regards to availabilty of ART, PREP, PEP etc, literacy, child marriages rates and other relevant information concerning the study area for the readers to connect the dots.

Since the intervention has definitely significant impact in PLHIV, authors can add a sub section at the end of discussion on relevance for policy making in resource constrained settings for HIV prevention and holistic care.
---

## [Editor Report · Decision Letter 3]

PGPH-D-24-02060R3

Impact of the DREAMS Partnership on adolescent girls and young women’s empowerment for HIV risk reduction: a mixed methods evaluation in Kenya

Dear Dr. Kemigisha,

Thank you for submitting your manuscript to PLOS Global Public Health. After careful consideration, we feel that it has merit but does not fully meet PLOS Global Public Health’s publication criteria as it currently stands. Therefore, we invite you to submit a revised version of the manuscript that addresses the points raised during the review process.

We look forward to receiving your revised manuscript.

Kind regards,

Nancy Angeline Gnanaselvam

Academic Editor

Additional Editor Comments (if provided):

Since the authors feel that the paper does not focus on impact among PLHIV, the relevance of paper in PGPH journal is doubtful. Authors are requested to elaborate on how the paper is in alignment with the journal which aims to address global inequities and creating impactful research in public health.

The title of the paper could be changed to " Impact of a multilevel programming and empowerment intervention for HIV risk reduction among adolescent girls and young women: A mixed methods evaluation in Kenya". DREAMS can be explained in the paper 
---

## [Editor Report · Decision Letter 4]

Impact of a multilevel HIV prevention programme on adolescent girls and young women’s empowerment for HIV risk reduction: a mixed methods evaluation in Kenya

PGPH-D-24-02060R4

Dear Dr Kemigisha,

We are pleased to inform you that your manuscript 'Impact of a multilevel HIV prevention programme on adolescent girls and young women’s empowerment for HIV risk reduction: a mixed methods evaluation in Kenya' has been provisionally accepted for publication in PLOS Global Public Health.

Best regards,

Nancy Angeline Gnanaselvam

Academic Editor

Authors have responded to the queries and revised the paper accordingly. The paper can be accepted for publication